

# The impact of hypertension on the prognosis of patients with hypertrophic cardiomyopathy: a single-center retrospective study

Ziqiong Wang[1,*], Yi Zheng[1,*], Haiyan Ruan[1,2], Liying Li[1], Muxin Zhang[1,3], Linjia Duan[1] and Sen He[1]

[1] Department of Cardiology, West China Hospital of Sichuan University, Chengdu, China
[2] Department of Cardiology, Hospital of Traditional Chinese Medicine, Shuangliu District, Chengdu, Sichuan, China
[3] Department of Cardiology, First People's Hospital, Longquanyi District, Chengdu, Sichuan, China
[*] These authors contributed equally to this work.

Corresponding author
Sen He, hesensubmit@163.com

## ABSTRACT

**Background**. Hypertrophic cardiomyopathy (HCM) and hypertension coexist fairly frequently in clinical practice. However, the evidence about the impact of hypertension on the prognosis of HCM is limited. The present study aims to investigate the impact of hypertension on the prognosis of HCM patients.

**Methods**. A total of 468 HCM patients were enrolled, and patients were divided into hypertension group (31.8%) and non-hypertension group (68.2%). The primary study endpoint was HCM-related death, consisting of heart failure (HF)-related death, stroke-related death and sudden cardiac death (SCD). Associations between hypertension and HCM-related death were analyzed by Cox regression models with the use of propensity score matching (PSM) as primary analysis.

**Results**. There were 55 HCM-related death during a median follow-up time of 4.6 years, and the mortality rate was 2.53 per 100 person years. Kaplan-Meier analysis based on the crude cohort or PSM cohort revealed no significant difference regarding the HCM-related death between the two groups. In the crude cohort, both univariable and multivariable Cox regression analysis indicated that hypertension was not significantly associated with HCM-related death with hazard ratios (HR) at 0.74 (95% CI [0.40–1.36], $p$ value: 0.329) and 0.77 (95% CI [0.35–1.71], $p$ value: 0.521), respectively. Similarly, no strong evidence for an association was observed between hypertension and HCM-related death in the PSM cohort with unadjusted HR at 0.90 (95% CI [0.34–2.41]; $p$ value: 0.838) and adjusted HR at 0.77 (95% CI [0.35–1.71]; $p$ value: 0.521), respectively. Other propensity score methods, including overlap weighting and inverse probability treatment weighting demonstrated similar results. Sensitivity analysis also indicated that the concomitant hypertension did not significantly increase the risk of HF-related death, stroke-related death or SCD in HCM patients.

**Conclusion**. HCM-related death did not significantly differ between hypertension and non-hypertension groups, suggesting a negative impact of hypertension on the clinical prognosis of HCM patients.

## INTRODUCTION

Hypertrophic cardiomyopathy (HCM) is an inherited heart disease with potential phenotypic heterogeneity in genetic variants, myocardial morphologies, symptoms, hemodynamics, cardiac systolic and diastolic function, as well as clinical prognosis (*Maron & Maron, 2013*; *Kitaoka, Kubo & Doi, 2020*). Previously, HCM was regarded as having a poor prognosis, including heart failure (HF), thromboembolic events, malignant arrhythmias, and mortality (*Maron et al., 2000*; *Maron, 2002*; *Guttmann et al., 2014*). During the last two decades, considering the application of contemporary treatment options, many patients managed to survive to normal or extended longevity with good quality of life, and the mortality directly attributable to HCM has been substantially reduced (*Maron, Maron & Rowin, 2017*). However, recent studies have shown the presence of other cardiac or noncardiac comorbidities might have a greater impact on survival than HCM itself in patients with HCM (*Wasserstrum et al., 2019*; *Maron et al., 2013*; *Sorajja et al., 2003*). Among these comorbidities, hypertension frequently coexists with HCM, and is present in 30–50% of HCM patients (*Wang, 2019*). It is well known that hypertension is one of the most prevalent cardiovascular diseases, affecting more than 1 billion individuals worldwide (*Mills, Stefanescu & He, 2020*). The co-existence of both conditions poses challenges during disease diagnosis and treatment. On one hand, hypertension per se can cause left ventricular hypertrophy due to hemodynamic overloading, which is characterized by concentric or eccentric hypertrophy, hyperdynamic left ventricular contraction and diastolic dysfunction (*Ganau et al., 1992*). On the other hand, the management of hypertension relies on antihypertensive drugs, some of which except beta-blockers may be contraindicated in HCM patients, especially in patients with obstructive HCM (*Ommen et al., 2020*). Previous studies have demonstrated that concomitant hypertension could affect cardiac structure in HCM patients, while the clinical outcomes, such as 5-year survival rate or cardiac death did not significantly differ between HCM patients with or without hypertension (*Luo et al., 2020*; *Deng et al., 2019*; *Aslam et al., 2010*). However, those studies had relatively small population, and mainly focused on all-cause mortality or cardiac mortality. The impact of hypertension on the prognosis of HCM patients, especially the HCM-related death is largely unknown. Herein, in the present study, we aimed to investigate the prognostic value of hypertension on HCM-related death in HCM patients.

## METHODS

### Study patients

This retrospective, single-center cohort study was performed at West China Hospital of Sichuan University, which a tertiary center located in Chengdu, China. From December 2008 to November 2018, we included 546 hospitalized patients with a discharge diagnosis of HCM in total. Baseline characteristics were collected from medical records by experienced physicians. Data entry was performed using the twice-entry method. The data would be entered into the database when the values of the two entries were same; otherwise, the raw data would be examined. After reviewing the medical records, nine patients were

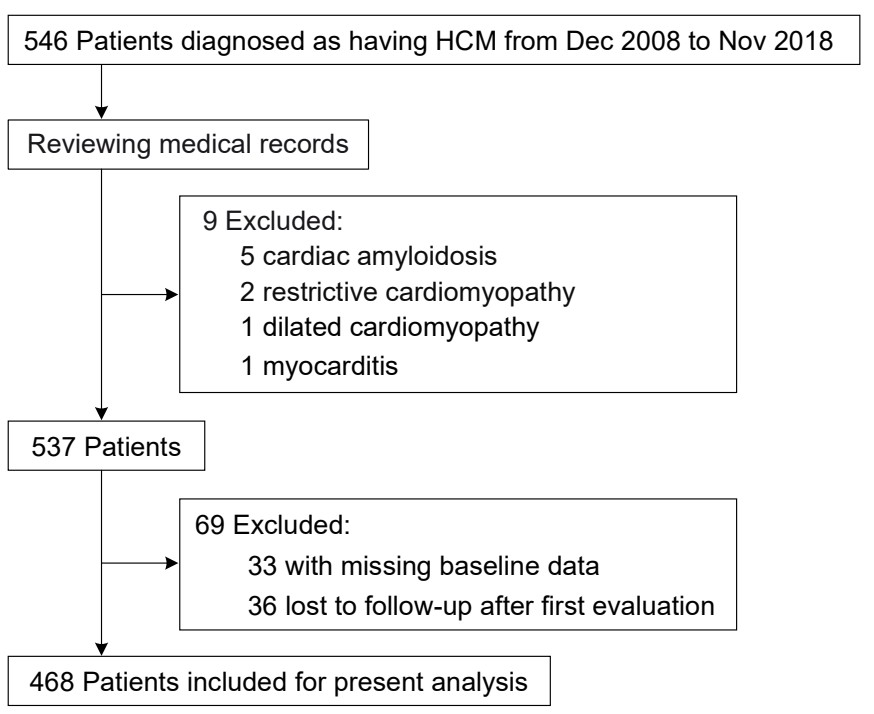

**Figure 1  Study flow diagram.**

excluded for other conditions that may cause cardiac hypertrophy, and 69 patients were excluded due to missing baseline data after the first evaluation or loss to follow-up (Fig. 1). Eventually, 468 adult patients in total were included for the present analysis. The Biomedical Research Ethics Committee of West China Hospital of Sichuan University approved the study (approval number: 2019-1147), and the study was conducted according to the principles of the Declaration of Helsinki. Because of the retrospective nature of the study, informed consent was waived. The study has been registered at the Chinese Clinical Trial Registry with a registration number of ChiCTR2000029352; the registration information can also be accessed via International Clinical Trials Registry Platform (https://trialsearch.who.int/default.aspx). Some other detailed information has been reported in the recently published study (*Wang, Zhao & He, 2021*).

## Diagnostic criteria

Diagnosis of HCM was made in accordance with the 2014 European Society of Cardiology guideline criteria (*Zamorano et al., 2014*), and it was principally supported by the echocardiographic finding of increased left ventricular (LV) wall thickness ($>= 15$ mm)—that was not solely explained by aberrant loading conditions. Each patient had $>=1$ of the following disease features of HCM (*Maron et al., 2013*; *Wang, 2019*): LV wall thickness $>=18$ mm ($n = 310$); systolic anterior motion (SAM) of mitral valve ($n = 146$); family history of HCM ($n = 40$); family history of sudden cardiac death (SCD) ($n = 15$); intervention of obstruction ($n = 52$); LV apical aneurysm ($n = 10$); asymmetric septal

hypertrophy ($n = 333$; IVS/LVPW $>= 1.5$); hyperdynamic LV function ($n = 208$; LV ejection fraction $>= 70\%$); non-long standing and refractory hypertension ($n = 468$; prior to admission, blood pressure was controlled within the target range).

The diagnosis of hypertension was made when the resting systolic blood pressure was $>= 140$ mm Hg and/or the diastolic blood pressure was $>= 90$ mm Hg, or when there was a history of antihypertensive drug usage (*Mancia et al., 2018*).

## Study outcome

Based on the previous studies (*Songsirisuk et al., 2019*), the study's outcome were described as a composite of three common types of HCM-related death and additional types of particular HCM-related death, such as: (1) HF-related death, which was defined as death preceded by signs and/or symptoms of HF lasting more than one hour, in the context of progressive cardiac decompensation more than a year before death (*Coats et al., 2013*); (2) stroke-related death, which occurred as a result of probable or proven ischemic stroke (*Haruki, Minami & Hagiwara, 2016*). Here we did not distinguish cardioembolic stroke from other ischemic subtypes in the present study; (3) SCD, which was defined as nocturnal death or witnessed sudden cardiac death within one hour of new symptoms in patients who previously had a relatively stable or uneventful clinical course (*Elliott et al., 2000*); (4) other specific HCM-related death, which was defined as death due to HCM, but not belonged to the aforementioned three conditions.

Follow-up was carried out *via* medical records or telephone contact with the patients themselves and/or referring relatives. All patients were followed from the first evaluation up to the endpoint or the most recent evaluation.

## Statistical analysis

Frequencies and descriptive statistics were used to summarize patient baseline characteristics for the research population.

Cox proportional-hazards regression models were used to estimate the association between hypertension and the study outcome. Initially, demographic factors, clinical factors, laboratory tests and therapies were included in a multivariable Cox regression model. propensity scores (PS) were developed to account for potential confounding by observed baseline characteristics given the observational nature of the current study (*Austin, 2011*). With PS methods, more potential confounders can be considered than with traditional regression techniques, which allows for the replacement of the whole set of baseline data with a single composite score. With the use of the multivariable logistic regression model, which took into account all the factors listed in Table 1, the propensities of each individual with hypertension were calculated. Associations between hypertension and the study outcome were then estimated by Cox regression models with the use of multiple propensity-score methods.

In the primary analysis, we used PS matching (PSM), in which 1:1 matching without replacement was carried out using a nearest neighbor matching algorithm with a fixed caliper width of 0.2. Additionally, in order to ensure the reproducibility of the analysis results by PSM, stabilized inverse probability treatment weighting (IPTW) (*Austin, 2016*)
**Table 1 Baseline characteristics.**

| Variable | Overall | Non-Hypertension | Hypertension | ASD |
|---|---|---|---|---|
| No. of patients (*n*) | 468 | 319 | 149 | |
| Gender, male | 215 (45.9) | 140 (43.9) | 75 (50.3) | 0.129 |
| Age (years) | 57.50 (46.00, 67.00) | 52.00 (41.50, 63.00) | 65.00 (57.00, 73.00) | 0.924 |
| Family history of HCM | 40 (8.5) | 33 (10.3) | 7 (4.7) | 0.215 |
| Family history of SCD | 15 (3.2) | 12 (3.8) | 3 (2.0) | 0.105 |
| NYHA34 | 162 (34.6) | 111 (34.8) | 51 (34.2) | 0.012 |
| Symptom | | | | |
| Dyspnea | 262 (56.0) | 181 (56.7) | 81 (54.4) | 0.048 |
| Chest pain | 249 (53.2) | 160 (50.2) | 89 (59.7) | 0.193 |
| Pre-/syncope | 146 (31.2) | 114 (35.7) | 32 (21.5) | 0.32 |
| Palpitation | 174 (37.2) | 123 (38.6) | 51 (34.2) | 0.09 |
| Heart rate | 72.00 (64.00, 80.00) | 72.00 (65.00, 80.00) | 72.00 (62.00, 81.00) | 0.091 |
| SBP | 120.00 (108.75, 135.00) | 116.00 (105.00, 127.50) | 136.00 (122.00, 150.00) | 1.012 |
| DBP | 70.50 (64.00, 80.00) | 70.00 (64.00, 80.00) | 78.00 (70.00, 85.00) | 0.559 |
| *Medical history* | | | | |
| Prior TE | 22 (4.7) | 7 (2.2) | 15 (10.1) | 0.333 |
| Vascular diseases | 37 (7.9) | 16 (5.0) | 21 (14.1) | 0.313 |
| Diabetes | 39 (8.3) | 15 (4.7) | 24 (16.1) | 0.38 |
| Atrial fibrillation | 83 (17.7) | 52 (16.3) | 31 (20.8) | 0.116 |
| *Therapy* | | | | |
| Beta blockers | 338 (72.2) | 233 (73.0) | 105 (70.5) | 0.057 |
| ACEI | 39 (8.3) | 23 (7.2) | 16 (10.7) | 0.124 |
| ARB | 52 (11.1) | 12 (3.8) | 40 (26.8) | 0.677 |
| Dihydropyridine | 41 (8.8) | 0 (0.0) | 41 (27.5) | 0.871 |
| Hydrochlorothiazide | 28 (6.0) | 12 (3.8) | 16 (10.7) | 0.272 |
| Aspirin | 85 (18.2) | 44 (13.8) | 41 (27.5) | 0.344 |
| Warfarin | 43 (9.2) | 28 (8.8) | 15 (10.1) | 0.044 |
| Statins | 134 (28.6) | 57 (17.9) | 77 (51.7) | 0.759 |
| Intervention of obstruction | | | | |
| None | 416 (88.9) | 281 (88.1) | 135 (90.6) | |
| Alcohol septal ablation | 45 (9.6) | 32 (10.0) | 13 (8.7) | 0.119 |
| Septal myectomy | 7 (1.5) | 6 (1.9) | 1 (0.7) | |
| Devices | | | | |
| None | 411 (87.8) | 275 (86.2) | 136 (91.3) | |
| Pacemaker | 21 (4.5) | 13 (4.1) | 8 (5.4) | 0.264 |
| ICD | 36 (7.7) | 31 (9.7) | 5 (3.4) | |
| *Hematological result* | | | | |
| Creatinine (μmol/L) | 80.00 (67.00, 94.17) | 77.00 (66.00, 90.00) | 85.00 (71.00, 101.90) | 0.141 |
| Uric acid (μmol/L) | 362.00 (299.08, 435.92) | 355.00 (291.70, 432.65) | 389.00 (316.00, 445.70) | 0.147 |
| TG (mmol/L) | 1.25 (0.93, 1.86) | 1.21 (0.90, 1.81) | 1.32 (0.99, 2.02) | 0.095 |
| HDL-C (mmol/L) | 1.27 (1.02, 1.55) | 1.26 (1.02, 1.52) | 1.30 (1.03, 1.63) | 0.175 |

**Table 1** (*continued*)

| Variable | Overall | Non-Hypertension | Hypertension | ASD |
|---|---|---|---|---|
| LDL-C (mmol/L) | 2.38 (1.81, 2.89) | 2.37 (1.81, 2.89) | 2.44 (1.80, 2.94) | 0.065 |
| *Echocardiographic data* | | | | |
| LVEDD (mm) | 43.00 (39.00, 46.00) | 42.00 (39.00, 46.00) | 44.00 (40.00, 48.00) | 0.166 |
| LA diameter (mm) | 40.00 (35.00, 45.00) | 40.00 (35.00, 45.00) | 41.00 (36.00, 45.00) | 0.058 |
| IVS (mm) | 19.00 (16.00, 22.00) | 19.00 (16.50, 22.00) | 19.00 (16.00, 21.00) | 0.192 |
| LVPW (mm) | 11.00 (10.00, 13.00) | 11.00 (9.00, 12.50) | 11.00 (10.00, 13.00) | 0.192 |
| IVS/LVPW | 1.67 (1.40, 2.10) | 1.70 (1.42, 2.15) | 1.60 (1.36, 1.91) | 0.299 |
| MWT (mm) | 19.00 (17.00, 22.00) | 19.00 (17.00, 22.00) | 19.00 (17.00, 21.00) | 0.184 |
| LVEF (%) | 69.00 (63.00, 72.00) | 69.00 (64.00, 73.00) | 68.00 (63.00, 71.00) | 0.065 |
| Resting LVOTG >= 30 mm Hg | 201 (42.9) | 126 (39.5) | 75 (50.3) | 0.219 |
| LV apical aneurysm | 10 (2.1) | 9 (2.8) | 1 (0.7) | 0.165 |
| SAM | 146 (31.2) | 108 (33.9) | 38 (25.5) | 0.184 |

**Notes.**

Values are median (IQR) or *n* (%).

ASD, absolute standardized differences; HCM, hypertrophic cardiomyopathy; SCD, sudden cardiac death; NYHA, New York Heart Association; SBP, systolic blood pressure; DBP, diastolic blood pressure; TE, thrombo-embolic event; ACE, angiotensin-converting enzyme inhibitor; ARB, angiotensin receptor blocker; ICD, implantable cardioverter defibrillator; TG, triglyceride; HDL-C, high density lipoprotein cholesterol; LDL-C, low density lipoprotein cholesterol; LVEDD, left ventricular end-diastolic dimension; LA, left atrial; IVS, interventricular septum; LVPW, left ventricular posterior wall; MWT, maximal left ventricular wall thickness; LVEF, left ventricular ejection fraction; LVOTG, left ventricular outflow tract gradient; LV, left ventricular; SAM, systolic anterior motion.

and overlap weighting (*Thomas, Li & Pencina, 2020*) were also carried out. The overall PS distributional curves and the absolute standardized differences (ASD) for each covariate (an ASD $\geq$ 0.10 indicates imbalance) were used to assess the covariate differences before and after PSM, as well as IPTW and overlap weighting (*Austin, 2016*). Then, we presented the Kaplan Meier curves and Cox models that used the above-mentioned PS, and we also showed the Cox model that included PS as an additional covariate. As the sensitivity analyses, we evaluated the relationship between hypertension and some specific HCM-related deaths, such as HF-related death, stroke-related death and SCD.

The statistical analyses were performed with the use of R software, version 4.1.0 (*R Core Team, 2021*). A two-sided *p* value of 0.050 was regarded as statistically significant for all statistical analyses.

# RESULTS

## Baseline Characteristics and study outcome

The study population consisted of 468 HCM patients, including 149 (31.8%) with hypertension and 319 (68.2%) without hypertension. HCM patients with hypertension were older, had lower prevalence of family history of HCM and SCD, had higher chest pain but lower pre-/syncope. More importantly, HCM patients with hypertension had a greater prevalence of prior thromboembolism, arial fibrillation and diabetes. Other drug applications except beta-blockers and warfarin were higher in hypertension group, while the intervention of obstruction and devices therapy were higher in non-hypertension group. In addition, HCM patients with hypertension tended to have larger LV end-diastolic diameter, higher prevalence of left ventricular outflow tract obstruction (LVOTO), lower prevalence of LV apical aneurysm and systolic anterior motion of the mitral valve. The

relatively thinner interventricular septum (IVS) and thicker left ventricular posterior wall (LVPW) resulted in a lower IVS/LVPW ratio in hypertension group. Other detailed information is shown in Table 1.

During a follow-up time of 2170.7 person-years (PYs) (median, 4.6 years; IQR, 2.1–6.8 years), a total of 55 HCM-related death occurred with a mortality rate of 2.53 (95% confidence interval [CI]: 1.87–3.19) per 100 PYs. The specific causes of deaths were as follows: 28 HF-related deaths, 10 stroke-related deaths, 15 SCDs, two HCM-related postoperative deaths.

## Bivariable analysis

In bivariable analysis (unweighted sample), the HCM-related death rates were 2.75 (95% CI [1.92–3.58]) and 2.06 (95% CI [0.99–3.13]) per 100 PYs in the non-hypertension and hypertension groups, respectively. Kaplan–Meier curves demonstrated that the cumulative HCM-related death rate did not significantly differ between the two groups (Fig. 2A). We used the function, namely cox.zph, from the package "survival" to test the proportional hazard assumption for hypertension, which was not violated ($p = 0.550$). Univariate Cox regression analysis indicated that there was no significant difference regarding HCM-related death between the two groups (HR = 0.74; 95% CI [0.40–1.36]; $p$ value: 0.329). After adjusting potential confounding factors, HR for HCM-related death in hypertension group *versus* non-hypertension group is 0.77 (95% CI [0.35–1.71], $p$ value: 0.521).

## Propensity score matching analysis

Results of multivariable logistic regression analysis that predicted hypertension are listed in Table 2, and the C-statistic of the model was 0.918. The distribution of PS in the hypertension and non-hypertension groups before matching is shown in Fig. 3A. The lesser overlap of PS curves of the two groups indicated a greater risk of confounding. With the use of PSM, 71 HCM patients with hypertension were matched with 71 HCM patients without hypertension. After PSM, PS curves for hypertension and non-hypertension groups were superimposed, indicating well-balanced baseline covariates between the two groups (Fig. 3B). Few baseline variables remained unbalanced with ASD ≥ 0.1 were added as covariates in the multivariable Cox regression analysis (Fig. 3E). Kaplan–Meier analysis also demonstrated that the cumulative HCM-related death rate did not significantly differ between the two groups after matching with log rank $p$ at 0.84 (Fig. 2B). Furthermore, in the primary analysis with PSM, both univariable and multivariable Cox regression analysis indicated that hypertension was not significantly associated with HCM-related death with HR at 0.90 (95% CI [0.34–2.41]; $p$ value: 0.838) and 0.77 (95% CI [0.35–1.71]; $p$ value: 0.521), respectively (Table 3).

## ITPW and overlap weighting analyses

After IPTW and overlap weighting, as compared to the crude cohort, the baseline differences between the hypertension group and non-hypertension group were largely reduced (Figs. 3C and 3D). The remained unbalanced variables were further adjusted in the multivariable Cox regression analysis as well (Fig. 3E). Kaplan–Meier curves indicated that the cumulative HCM-related death was comparable between the two groups after IPTW and overlap

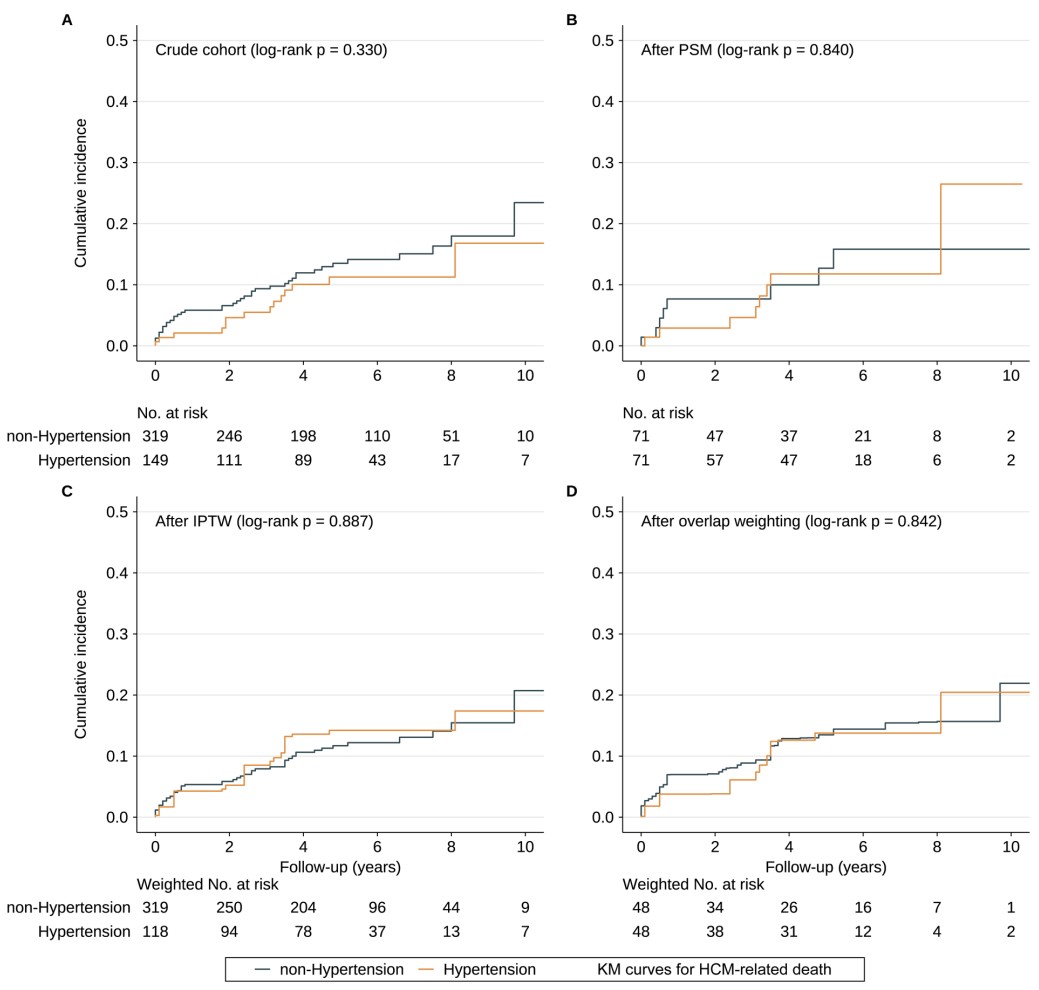

**Figure 2 Cumulative incidence of HCM-related death.** (A–D) The cumulative incidence of HCM-related death in the subjects with or without hypertension in the crude sample, PSM sample, IPTW sample and overlap weighting sample. Abbreviations as in Table 1.

weighting with log rank *p* at 0.887 and 0.842, respectively (Figs. 2C and 2D). No strong evidence for an association was observed between hypertension and HCM-related death in both univariate and multivariate Cox regression models in IPTW and overlap weighting cohort (Table 3).

## Sensitivity analysis

As a sensitivity analysis, we explored the association between hypertension and some specific HCM-related death, which included HF-related death, stroke-related death and SCD. In crude analysis, no significant difference was found between the two groups for HF-related death (HR = 0.71; 95% CI [0.30–1.67]; *p* value: 0.434), stroke-related death (HR = 1.55; 95% CI [0.44–5.51]; *p* value: 0.498) and SCD (HR = 0.53; 95% CI [0.15–1.89]; *p* value: 0.330). In the primary analysis with PSM, unadjusted HRs of hypertension for HF-related death, stroke-related death and SCD were 0.87 (95% CI [0.22–3.49]; *p* value:

**Table 2  Multivariable logistic regression model predicting hypertension.**

| Variable | Changes | Beta coefficient | Std.error | p value |
|---|---|---|---|---|
| (Intercept) | | −19.035 | 4.151 | 0.000 |
| Gender, male | Female vs male | 0.320 | 0.374 | 0.392 |
| Age (years) | Per 1-unit increment | 0.058 | 0.015 | 0.000 |
| Family history of HCM | Yes vs no | −0.162 | 0.619 | 0.794 |
| Family history of SCD | Yes vs no | −1.413 | 1.156 | 0.222 |
| NYHA34 | Yes vs no | −0.222 | 0.355 | 0.532 |
| Dyspnea | Yes vs no | 0.279 | 0.355 | 0.432 |
| Chest pain | Yes vs no | 0.515 | 0.338 | 0.128 |
| Pre-/syncope | Yes vs no | −0.418 | 0.389 | 0.282 |
| Palpitation | Yes vs no | 0.169 | 0.340 | 0.619 |
| Heart rate | Per 1-unit increment | 0.003 | 0.012 | 0.781 |
| SBP | Per 1-unit increment | 0.038 | 0.011 | 0.001 |
| DBP | Per 1-unit increment | 0.027 | 0.018 | 0.136 |
| Prior TE | Yes vs no | 1.163 | 0.811 | 0.152 |
| Vascular diseases | Yes vs no | −0.180 | 0.603 | 0.766 |
| Diabetes | Yes vs no | 0.570 | 0.546 | 0.297 |
| Atrial fibrillation | Yes vs no | −0.022 | 0.551 | 0.967 |
| Beta blockers | Yes vs no | −0.022 | 0.402 | 0.957 |
| ACEI | Yes vs no | 0.262 | 0.524 | 0.618 |
| ARB | Yes vs no | 1.906 | 0.549 | 0.001 |
| Dihydropyridine | Yes vs no | 18.708 | 853.451 | 0.983 |
| Hydrochlorothiazide | Yes vs no | 0.968 | 0.640 | 0.130 |
| Aspirin | Yes vs no | −0.285 | 0.448 | 0.524 |
| Warfarin | Yes vs no | 0.020 | 0.634 | 0.974 |
| Statins | Yes vs no | 0.487 | 0.427 | 0.254 |
| Intervention of obstruction | | | | |
| None | | Ref. | | |
| Alcohol septal ablation | | 0.036 | 0.574 | 0.949 |
| Septal myectomy | | −0.535 | 1.407 | 0.704 |
| Devices | | | | |
| None | | Ref. | | |
| Pacemaker | | −0.709 | 1.058 | 0.503 |
| ICD | | −0.571 | 0.753 | 0.448 |
| Creatinine (μmol/L) | Per 1-unit increment | 0.000 | 0.002 | 0.976 |
| Uric acid (μmol/L) | Per 1-unit increment | 0.005 | 0.002 | 0.004 |
| TG (mmol/L) | Per 1-unit increment | 0.131 | 0.143 | 0.358 |
| HDL-C (mmol/L) | Per 1-unit increment | 0.145 | 0.403 | 0.719 |
| LDL-C (mmol/L) | per 1-unit increment | −0.046 | 0.202 | 0.818 |
| LVEDD (mm) | Per 1-unit increment | 0.022 | 0.030 | 0.463 |
| LA diameter (mm) | Per 1-unit increment | −0.014 | 0.025 | 0.572 |
| IVS (mm) | Per 1-unit increment | −0.021 | 0.316 | 0.948 |

**Table 2** (*continued*)

| Variable | Changes | Beta coefficient | Std.error | *p* value |
|----------|---------|------------------|-----------|-----------|
| LVPW (mm) | Per 1-unit increment | 0.223 | 0.176 | 0.206 |
| IVS/LVPW | Per 1-unit increment | 1.052 | 1.228 | 0.392 |
| MWT (mm) | Per 1-unit increment | −0.073 | 0.352 | 0.835 |
| LVEF (%) | Per 1-unit increment | 0.019 | 0.022 | 0.404 |
| Resting LVOTG >= 30 mm Hg | yes vs no | 1.235 | 0.450 | 0.006 |
| LV apical aneurysm | Yes vs no | −0.651 | 1.368 | 0.634 |
| SAM | Yes vs no | −0.685 | 0.467 | 0.142 |

**Notes.**

Individual propensities of subjects with hypertension were estimated with the use of a multivariable logistic regression model that included all the variables in the table. C-index = 0.918, and abbreviations as in Table 1.

0.846), 0.96 (95% CI [0.13–6.89]; *p* value: 0.967) and 0.91 (95% CI [0.13–6.49]; *p* value: 0.926), respectively. The results remained unchanged with further adjustment of age and sex in the PSM cohort. Additionally, multivariable analysis and other PS analyses unanimously demonstrated that hypertension was not significantly associated with those specific HCM-related death (Table 4).

# DISCUSSION

The present study compared two groups of HCM patients to examine the effect of coexisting hypertension on the prognosis of HCM. The cumulative HCM-related death was not significantly different between the non-hypertension group and hypertension group. Cox regression analysis further indicated that hypertension was not a significant risk factor for HCM-related death, including HF-related death, stroke-related death or SCD in HCM patients.

Consistent with previous studies (*Luo et al., 2020*; *Deng et al., 2019*; *Aslam et al., 2010*), the clinical features of HCM patients with hypertension in the present study were characterized by older age, higher application of other antihypertensive drugs except *β* blockers, such as renin-angiotensin system inhibitors (RASIs), dihydropyridine calcium channel antagonists (DHP-CCBs) and hydrochlorothiazide. Vasodilators were the most effective and tolerable drugs for blood pressure control but could exacerbate LVOTO. Historical studies indicated that the presence of LVOTO did not differ between HCM patients with hypertension or those without hypertension (*Luo et al., 2020*; *Aslam et al., 2010*). While the prevalence of LVOTO was much higher in the hypertension group of our study. In a follow-up study, *Argulian et al. (2013)* suggested that reducing the use of RASIs or DHP-CCBs and increasing the use of *β* blockers could decrease LVOTO and improve the cardiac function in HCM patients. Those comorbidities, including prior thromboembolism, vascular diseases, diabetes and atrial fibrillation were more common in those with hypertension, and thus the application of anti-platelet and statins drug therapy was higher in the hypertension group. In addition, collaborated with previous study (*Luo et al., 2020*), the prevalence of pre-/syncope is also much lower in HCM patients with hypertension, which might partially be explained by a greater blood pressure "reserve" for maintenance of consciousness at onset of syncope due to higher baseline blood pressure

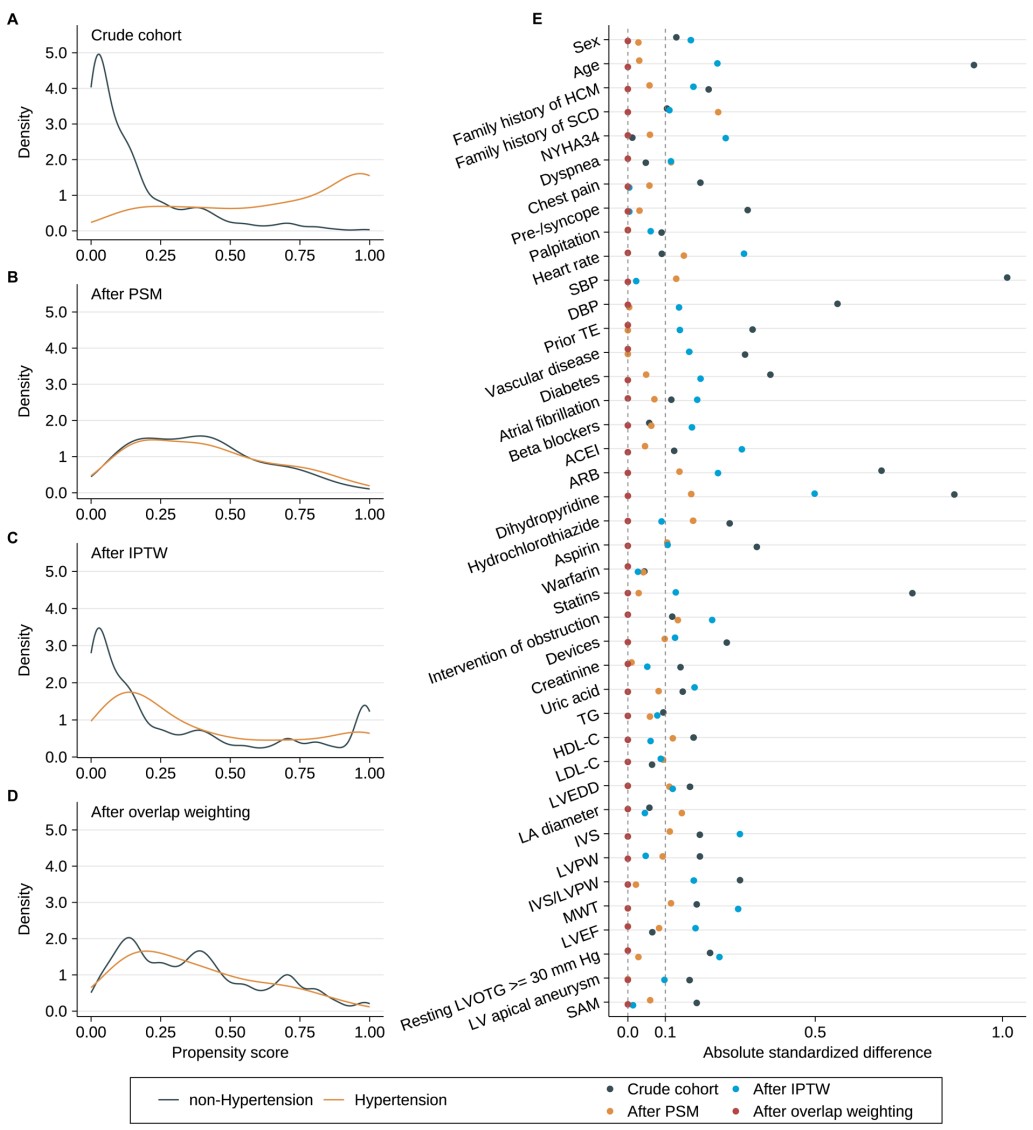

**Figure 3 Propensity scores distributional overlap and absolute standardized differences in the subjects with or without hypertension.** (A–D) PS distributions between the subjects with or without hypertension in the crude sample, PSM sample, IPTW sample and overlap weighting sample. For intervals along the $x$-axis, the area under the probability density curve represents the probability of those PSs, and smoothing was *via* the kernel density estimate. Greater overlap of PS curves of the two groups indicates a lesser risk of confounding. (E) ASD in the subjects stratified by hypertension. The dashed line indicates greater than 0.1 imbalance between the variable's value, which is a commonly used metric of significant imbalance. Abbreviations as in Table 1.

in hypertensive patients. The elevated basal blood pressure could counteract the drop in blood pressure caused by abnormal reflex control of vasculature, thereby reducing the incidence of syncope (*Luo et al., 2020*).

Another difference between the two groups was the cardiac structure. HCM patients with hypertension had relatively thinner IVS but thicker LVPW, and thus a lower IVS/LVPW

**Table 3  Associations between hypertension and HCM-related death.**

| Analysis | HCM-related death |
|---|---|
| No. of events/no. of patients at risk (%)[a] | |
|     Non-Hypertension | 41/319 (12.85%) |
|     Hypertension | 14/149 (9.40%) |
| Crude analysis | 0.74 (0.40–1.36), 0.329 |
| Multivariable analysis | 0.77 (0.35–1.71), 0.521[b] |
| Propensity-score analyses | |
|     With PSM (univariable) | 0.90 (0.34–2.41), 0.838 |
|     With PSM (multivariable) | 0.77 (0.35–1.71), 0.521[c] |
|     With IPTW (univariable) | 1.10 (0.47–2.56), 0.834 |
|     With IPTW (multivariable) | 1.35 (0.60–3.01), 0.467[d] |
|     With overlap weighting | 0.92 (0.40–2.11), 0.852 |
|     Adjusted for PS | 0.93 (0.40–2.19), 0.874[e] |

**Notes.**

Values are $n$ (%) or HRs (95% CI) with $p$ values.

[a]Binary event rates.

[b]Adjustment for imbalance variables, including sex, age, family history of HCM, chest pain, pre/syncope, SBP, DBP, prior TE, vascular disease, diabetes, atrial fibrillation, ACEI, ARB, dihydropyridine, hydrochlorothiazide, aspirin, statins, intervention of obstruction, devices, creatinine, uric acid, HDL-C, LVEDD, IVS, LVPW, IVS/LVPW, MWT, Resting LVOTG >= 30 mm Hg, LV apical aneurysm and SAM.

[c]Adjustment for imbalance variables after PSM, including family history of SCD, dyspnea, heart rate, SBP, ARB, dihydropyridine, hydrochlorothiazide, aspirin, intervention of obstruction, HDL-C, LVEDD, LA diameter, IVS and MWT.

[d]Adjustment for imbalanced variables after IPTW, including sex, age, family history of HCM, family history of SCD, NYHA34, dyspnea, heart rate, DBP, prior TE, vascular disease, diabetes, atrial fibrillation, beta blockers, ACEI, ARB, dihydropyridine, aspirin, statins, intervention of obstruction, devices, uric acid, LVEDD, IVS, IVS/LVPW, MWT, LVEF and Resting LVOTG >= 30 mm Hg.

[e]Only adjustment for PS.

These above-mentioned multivariable models were adjusted for the covariates which were found to be unbalanced even with the PSM approach.

PSM, propensity score matching; IPTW, inverse probability treatment weighting; PS, propensity score; HRs=HR, hazard ratios; CI, confidence interval, and other abbreviations as in Table 1.

ratio, representing a more concentric left ventricular hypertrophy (LVH). Both HCM and hypertension can lead to LVH with different mechanisms. LVH in HCM is caused by sarcomere protein mutations. Currently, there are more than 1,400 different mutations in at least 11 genes encoding cardiac sarcomere proteins have been identified, with gene MYBPC3 (cardiac myosin-binding protein C) and MYH7 (beta-myosin heavy chain) being most frequently involved (*Marian & Braunwald, 2017*; *Sabater-Molina et al., 2018*). Factors influencing LVH in patients with hypertension include the level, duration, and rapidity of onset of the increased afterload, the volume load, neurohumoral mechanisms and some genetic factors (*Aronow, 2017*). The pattern of LVH in hypertension is expected to be more concentric than eccentric as in HCM, but both patterns of hypertrophy have been described (*Cuspidi et al., 2012*). Therefore, the differentiation between HCM and hypertension represents a diagnostic challenge. No hypertension medical history or low level of elevated blood pressure, family history of HCM and/or SCD support HCM as the cause of LVH. The detection of dynamic LVOTO due to the SAM of the mitral valve and mitral-septal contact at rest or exertion, specific apical or apical-mild hypertrophy sparing the base also suggest the diagnosis of HCM. In addition, cardiac magnetic resonance (CMR) assessing the parameters of global longitudinal strain and late gadolinium enhancement

**Table 4** Associations between hypertension and some specific HCM-related death.

| Analysis | HF-related death | Stroke-related death | SCD |
|---|---|---|---|
| No. of events/no. of patients at risk (%)[a] | | | |
|     Non-Hypertension | 21/319 (6.58%) | 6/319 (1.88%) | 12/319 (3.76%) |
|     Hypertension | 7/149 (4.70%) | 4/149 (2.68%) | 3/149 (2.01%) |
| Crude analysis | 0.71 (0.30–1.67), 0.434 | 1.55 (0.44–5.51), 0.498 | 0.53 (0.15–1.89), 0.330 |
| Multivariable analysis[b] | 0.59 (0.24–1.45), 0.246 | 0.84 (0.21–3.33), 0.805 | 0.42 (0.11–1.54), 0.189 |
| Propensity-score analyses | | | |
|     With PSM (univariable) | 0.87 (0.22–3.49), 0.846 | 0.96 (0.13–6.89), 0.967 | 0.91 (0.13–6.49), 0.926 |
|     With PSM (multivariable)[b] | 0.99 (0.24–3.99), 0.983 | 1.14 (0.15–8.50), 0.898 | 0.95 (0.13–6.76), 0.959 |
|     With IPTW (univariable) | 0.87 (0.22–3.49), 0.846 | 0.96 (0.13–6.89), 0.967 | 0.91 (0.13–6.49), 0.926 |
|     With IPTW (multivariable)[b] | 0.99 (0.24–3.99), 0.983 | 1.14 (0.15–8.50), 0.898 | 0.95 (0.13–6.76), 0.959 |
|     With overlap weighting | 1.10 (0.34–3.51), 0.878 | 1.03 (0.15–7.07), 0.975 | 0.72 (0.17–3.07), 0.662 |
|     Adjusted for PS[c] | 1.08 (0.34–3.51), 0.892 | 1.13 (0.16–8.05), 0.904 | 0.62 (0.11–3.57), 0.594 |

Notes.

Values are $n$ (%) or HRs (95% CI) with $p$ values.

[a]Binary event rates.

[b]Adjustment for sex, age.

[c]Only adjustment for PS.

Abbreviations as in Tables 1 and 3.

(LGE) is also recommended to further differentiate the causes of LVH (*Neisius et al., 2019*; *Burrage & Ferreira, 2020*).

There is limited clinical evidence regarding the impact of hypertension on the clinical outcomes of HCM. *Luo et al. (2020)* demonstrated that the 5-year survival rate tended to be poorer in HCM patients with hypertension, but the results was not statistically significant. *Aslam et al. (2010)* reported no significant difference in terms of congestive HF, cardiac rest, myocardial infarction and SCD between HCM patients with hypertension or without. Besides, *Maron et al. (2003)* and *Maron et al. (2013)* also demonstrated that the history of hypertension had no significant influence on the survival or HCM-related death in their previous studies. In our study, we also found that there was no significant association between hypertension and HCM-related death. Although the constituent ratio of the causes of death slightly differed between the two groups. For example, HF-related death and SCD were more pronounced in patients with HCM alone, while stroke-related death was more pronounced in patients with HCM and hypertension. However, sensitivity analysis revealed that hypertension did not significantly increase the risk of HF-related death, stroke-related death or SCD in HCM patients. Despite that HCM patients with hypertension were older and had more comorbidities, corresponding therapeutic approaches by antihypertensive drugs, anti-platelet drugs and statins were given to patients with high-risk profiles. Therefore, the heart function at baseline as reflected by the proportion of New York Heart Association 3/4 and the value of LV ejection fraction was comparable between the two groups, which may be related to the negative association.

To some extent, patients with HCM and hypertension present clinicians with challenging therapeutic dilemma. It is well known that hypertension is one of the most prevalent cardiovascular diseases, causing a huge disease burden due to increased risk of ischemic

heart disease, HF, stroke and cardiovascular death (*Mills, Stefanescu & He, 2020*). HCM guideline recommended that target blood pressure should be in keeping with primary prevention guidelines (*Ommen et al., 2020*). Beta-blockers and verapamil/diltiazem would be the first line therapy for patients with HCM and hypertension to relieve the obstruction and control blood pressure simultaneously. However, the management of hypertension often requires other kinds of antihypertensive drugs in combination to achieve the target blood pressure goal, and thus minimalize the risk of cardiovascular and cerebrovascular events. In this condition, personalized therapeutic approaches to balance the management of HCM symptoms and blood pressure should be emphasized.

There were several limitations in the present study. Firstly, this was a single-center, retrospective study, with a limited sample size, so findings might not be generalized. But our sample size was much larger than previous studies (*Luo et al., 2020*; *Deng et al., 2019*; *Aslam et al., 2010*). Secondly, the diagnosis of HCM was based on the medical history and echocardiography, CMR was not widely applied for all patients due to the cost-effectiveness. Thirdly, the data of serial blood pressure monitoring was lacking. The management of hypertension and the achievement of targeted blood pressure are expected to mediate the prognostic effect of hypertension on HCM patients. Fourthly, although we have considered the potential confounding factors as much as possible, some other well-established predictors, such as ventricular arrhythmia, B type natriuretic peptide and cardiac biomarkers, were not included for analysis due to incomplete data, which might reduce the strength of the current finding. Further large-scale studies based on multi-centers are encouraged to investigate the prognostic value of hypertension in HCM.

## CONCLUSION

The clinical features of HCM patients with hypertension included older age, higher prevalence of other cardiovascular diseases and diabetes, and thus more intensive drug therapy was needed. HCM patients suffering from hypertension had relatively lower IVS/LVPW ratio, higher prevalence of LVOTO, but the symptom of pre-/syncope and the requirement of obstruction intervention or device implantation did not significantly increase. No significant association between hypertension and HCM-related death, including HF-related death, stroke-related death and SCD, was found. Prospective, multicenter-based studies with large samples are warranted to further illustrate the prognostic value of hypertension in HCM.

### Funding

This study was supported by the Sichuan Science and Technology Program, China (Grant No. 2022YFS0186), and the National Natural Science Foundation of China (Grant number: 81600299). The funders had no role in study design, data collection and analysis, decision to publish, or preparation of the manuscript.

## Grant Disclosures

The following grant information was disclosed by the authors:
Sichuan Science and Technology Program, China: 2022YFS0186.
National Natural Science Foundation of China: 81600299.

## Competing Interests

The authors declare there are no competing interests.

## Author Contributions

- Ziqiong Wang conceived and designed the experiments, analyzed the data, prepared figures and/or tables, authored or reviewed drafts of the article, and approved the final draft.
- Yi Zheng conceived and designed the experiments, analyzed the data, prepared figures and/or tables, authored or reviewed drafts of the article, and approved the final draft.
- Haiyan Ruan analyzed the data, prepared figures and/or tables, and approved the final draft.
- Liying Li analyzed the data, prepared figures and/or tables, and approved the final draft.
- Muxin Zhang analyzed the data, authored or reviewed drafts of the article, and approved the final draft.
- Linjia Duan analyzed the data, authored or reviewed drafts of the article, and approved the final draft.
- Sen He conceived and designed the experiments, analyzed the data, authored or reviewed drafts of the article, and approved the final draft.

## Human Ethics

The following information was supplied relating to ethical approvals (*i.e.*, approving body and any reference numbers):

The study was conducted according to the principles of the Declaration of Helsinki, and was approved by the Biomedical Research Ethics Committee, West China Hospital of Sichuan University (approval number: 2019-1147).

## Data Availability

The raw data is available in the Supplemental Files.

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
