# Peer review of "The impact of hypertension on the prognosis of patients with hypertrophic cardiomyopathy: a single-center retrospective study"

_PeerJ, doi:10.7717/peerj.14614_

## Round 0.1 · original submission · Minor Revisions

Dear authors,

Please reply point by point to the reviewers' comments.

Thanks!

Reviewer 1 ·

Basic reporting

Dear authors,
The article is well-written and rich in interesting content. English is clear and technical language is used.
The introduction is synthetic and smooth to read. I very much appreciated it. However, in my opinion, you could add some basic clinical information about hypertension in this section. Appropriate references have been reported.
All the figures are relevant to the content of the article, and their descriptions are clear. No further correction is needed.

Experimental design

The text is linear and the whole content clearly defines the research question. The study adds much information to the literature to fill that gap. Rigorous investigations have been performed and conformed to an ethical standard. Methods are described with sufficient information. I just suggest you insert a reference to whether the cut-off points used to diagnose hypertension were by any previous study (lines 90-92).

Validity of the findings

In my opinion, the results are well stated and exposed. Although the statistical methods are used correctly and several pieces of information were provided, it is not clear whether the Cox proportional-hazards regression model assumptions were met.
Discussion and conclusions are linked to the original research question and hypothesis.

Reviewer 2 ·

Basic reporting

The author’s study tries to investigate the effect of coexisting existence of hypertension on the prognosis of Hypertrophic cardiomyopathy (HCM) by comparing two groups of HCM.
The manuscript is generally well-written and easy to read; a slight spell-check is required.


Abstract

keywords usually should be different from that used in the main title.

Introduction

The literature on the subject could be improved, perhaps indicating the incidence of this condition and the anatomopathological features of the heart.

Experimental design

Methods

The methods section is sufficiently well described.

Validity of the findings

The results and discussion section are quite clear and organised.
The parameters considered are well presented.

---

## Round 0.2 · accepted · Accept

Dear authors

Thank you for your replies!